# Distinct Immunophenotypes in the DNA Index-Based Stratification of Pediatric B-Cell Acute Lymphoblastic Leukemia

**DOI:** 10.3390/cancers16213585

**Published:** 2024-10-24

**Authors:** Myriam Campos-Aguilar, Wilfrido David Tapia-Sánchez, Alberto Daniel Saucedo-Campos, Carlos Leonardo Duarte-Martínez, Sandra Olivas-Quintero, Almarosa Ruiz-Ochoa, Adolfo Rene Méndez-Cruz, Julia Reyes-Reali, María Isabel Mendoza-Ramos, Rafael Jimenez-Flores, Glustein Pozo-Molina, Elias Piedra-Ibarra, Maria Eugenia Vega Hernandez, Leopoldo Santos-Argumedo, Victor Hugo Rosales-García, Alberto Ponciano-Gómez

**Affiliations:** 1Laboratorio de Inmunología (UMF), Facultad de Estudios Superiores Iztacala, Universidad Nacional Autónoma de México, Los Barrios N° 1, Los Reyes Iztacala, Tlalnepantla 54090, Estado de México, Mexico; 2Diagnóstico Molecular de Leucemias y Terapia Celular (DILETEC), Basiliso Romo Anguiano 124, Industrial, Gustavo A. Madero, Mexico City 07800, Mexico; 3Departamento de Ciencias de la Salud Culiacán, Universidad Autónoma de Occidente, Culiacan 80020, Sinaloa, Mexico; sandra.olivas@uadeo.mx; 4Laboratorio de Genética y Oncología Molecular, Carrera de Médico Cirujano, Facultad de Estudios Superiores Iztacala, Universidad Nacional Autónoma de México, Los Barrios N° 1, Los Reyes Iztacala, Tlalnepantla 54090, Estado de México, Mexico; 5Fisiología Vegetal (UBIPRO), Facultad de Estudios Superiores Iztacala, Universidad Nacional Autónoma de México, Los Barrios N° 1, Los Reyes Iztacala, Tlalnepantla 54090, Estado de México, Mexico; 6CITOLAB, Av. Ejercito Nacional 613, Col Granada, Miguel Hidalgo, Mexico City 11520, Mexico; 7Departamento de Biomedicina Molecular, Centro de Investigación y de Estudios Avanzados del Instituto Politécnico Nacional (CINVESTAV-IPN), Mexico City 07360, Mexico; 8Centro de Investigación Sobre el Envejecimiento, Centro de Investigación y de Estudios Avanzados del Instituto Politécnico Nacional (CINVESTAV-IPN), Mexico City 07360, Mexico; 9Laboratorios Nacionales de Servicios Experimentales, Centro de Investigación y de Estudios Avanzados del Instituto Politécnico Nacional (CINVESTAV-IPN), Mexico City 14330, Mexico

**Keywords:** DNA index, immunophenotyping, B-LLA, hypoploidy, prognostic stratification, S phase, CD34, CD22, HLA-DR

## Abstract

B-cell acute lymphoblastic leukemia is a complex blood cancer that primarily affects children. Predicting treatment response is challenging due to the variability of the disease. In our study, we classified patients into three groups based on their DNA content (hyperploid, normoploid, and hypoploid). We then analyzed the markers present on leukemia cells using flow cytometry to identify potential differences between these groups. We found that hypoploid patients had a lower percentage of cells in the S phase of the cell cycle and showed higher levels of the HLA-DR marker, along with the co-expression of CD34 and CD22. These findings provide insights that could contribute to a better understanding and management of this disease in the future.

## 1. Introduction

B-cell acute lymphoblastic leukemia (B-ALL) is the most common hematologic malignancy in children, representing a significant challenge in terms of diagnosis and treatment due to its marked biological and clinical heterogeneity [1]. This heterogeneity leads to a wide variability in treatment outcomes, which has driven the need to develop more precise risk stratification methods to better tailor therapies to the individual characteristics of patients [2]. In this context, the DNA index (DNAi) has emerged as an essential tool for classifying B-ALL patients into various prognostic categories based on their DNA content, distinguishing patients into hyperploid, normoploid, and hypoploid groups, each associated with different clinical outcomes [3,4,5,6].

The DNAi, which measures DNA content in leukemic cells, has proven to be a strong predictor of outcomes in B-ALL. For instance, patients with a DNA index indicating hypoploidy often face a more uncertain or poor prognosis compared to those with normoploidy or hyperploidy [4,5,6]. Further studies have demonstrated that hypoploidy is associated with a higher risk of relapse and lower overall survival, underscoring the importance of precise patient stratification to guide therapeutic decisions [7].

In addition to the DNAi, immunophenotypic markers have been widely studied as additional prognostic indicators in B-ALL. Flow cytometry, a key technique in evaluating these markers, has enabled the detailed characterization of leukemic cells, identifying specific expression patterns that can correlate with ploidy categories and, consequently, patient prognosis [3,8]. For example, the expression of CD10, CD22, and other markers has been particularly relevant in previous studies, suggesting that these markers could provide additional information about disease aggressiveness and treatment response [8,9].

However, the relationship between DNAi and specific immunophenotypic profiles remains an active area of research. Some studies have suggested that patients with ploidy alterations may present unique immunophenotypic characteristics that could contribute to their poorer prognosis, although the data have not always been conclusive [9]. This variability in results highlights the need for larger and more detailed studies to explore these possible associations more thoroughly [7].

This study focuses on exploring whether specific immunophenotypic markers are associated with each ploidy group based on the DNA index and whether these associations could be linked to prognostic categories. Using flow cytometry, we analyzed immunophenotypic markers in a cohort of B-ALL patients, integrating these data with DNA index measurements to stratify patients into specific risk categories. This approach has the potential to improve the accuracy of patient classification, providing new insights into the possible prognostic implications of these findings.

Identifying subgroups with particular immunophenotypic characteristics could have important clinical implications, especially regarding the personalization of treatment. Although our study does not focus on investigating new therapies, the findings on immunophenotypic differences between subgroups could influence the choice of targeted therapies and the development of new therapeutic approaches in the future. In particular, understanding the immunophenotypic differences between B-ALL subgroups based on the DNA index could provide valuable information for risk stratification and prognosis prediction, paving the way for future research that optimizes care for these complex patients.

## 2. Materials and Methods

### 2.1. Population Acquisition and Sample Selection

From 2019 to 2020, at the Regional Hospital of Tlalnepantla, State of Mexico, ISSEMyM, we identified 100 pediatric patients diagnosed with acute lymphoblastic leukemia (ALL) using morphological analysis, specifically utilizing Wright–Giemsa staining for the evaluation of peripheral blood and bone marrow smears. The inclusion criteria were a confirmed diagnosis of B-cell ALL (B-ALL) via immunophenotyping, the exclusion of patients with pre-existing genetic syndromes such as Down syndrome or other conditions involving chromosomal number changes, and the availability of an adequate bone marrow sample for DNA index (DNAi) analysis. Of the identified patients, 64 met the inclusion criteria and were selected for the study.

### 2.2. Immunophenotyping

Bone marrow aspirates obtained at the time of diagnosis were used for immunophenotyping, and to determine absolute counts of lymphocytes, monocytes, and granulocytes using a Cell-Dyn EMERALD Hematology Analyzer (Abbott, Chicago, IL, USA). The markers analyzed included CD2, CD3, CD4, CD5, CD7, CD8, CD10, CD11b, CD13, CD14, CD15, CD16, CD19, CD20, CD22, CD34, CD33, CD41a, CD45, CD56, CD71, CD79a, CD117, CD133, CD23ab, HLA-DR, TdT, IgM, and MPOx. These markers were stained using a comprehensive panel designed for leukemia diagnosis, and the staining details, including clones, fluorochromes, and dilutions, are provided in Appendix A. The stained specimens were then assessed using a CytoFLEX flow cytometer (Beckman Coulter, Morrisville, NC, USA), and blast cells were identified based on CD45+ and side-scatter (SSC) staining and analyzed using Kaluza software 2.1 (Beckman Coulter, Morrisville, NC, USA).

### 2.3. DNA Index Determination

To assess the DNA index (DNAi), total bone marrow white blood cells were fixed overnight in 70% ethanol, followed by centrifugation at 200× *g* for 5 min at 4 °C. After discarding the ethanol, cells were resuspended in PBS, centrifuged again, and then stained with a DAPI staining solution (100 µg/mL) for 30 min in the dark. The DNA content of 20,000 cells per sample was measured using a CytoFLEX cytometer (Beckman Coulter, Morrisville, NC, USA). ModFit LT software version 6.0 (Verity Software House, Topsham, ME, USA) was used to analyze DNA content and cell-cycle parameters. Only samples with a coefficient of variation below 5% were considered.

### 2.4. Stratification of Patient Samples into Prognostic Categories

Patient samples were stratified into prognostic categories based on their DNA index values, following established criteria in the literature [10]. The DNAi was calculated from the total DNA content of leukemic cells, with a value of 1 assigned to those with normal DNA content. Patients with a DNAi > 1.16 were classified as having a good prognosis (hyperploidy, *n* = 20), those with a DNAi between 1 and 1.16 were classified as having a standard prognosis (normoploidy, n = 30), and patients with a DNAi < 1 were classified as having a poor prognosis (hypoploidy, n = 14) (Appendix A).

This stratification was further supported by considering initial clinical features, such as white blood cell counts and the presence of extramedullary disease, according to guidelines from the National Cancer Institute (NCI). These clinical features were used in conjunction with the DNAi to provide a more comprehensive assessment of prognosis, reflecting the combined impact of genetic and clinical factors.

### 2.5. Principal Component Analysis

Principal Component Analysis (PCA) was performed using the FactoMineR (http://factominer.free.fr/) and factoextra R (https://www.r-project.org/) packages to identify specific patterns and correlations within our dataset. The analysis focused on significant variables among patient groups, particularly considering prognosis and ploidy status. The dataset included variables such as immunophenotypic marker positivity, overall cell percentages, cell-cycle phase distribution from DNA index analysis, and cell-type percentages determined by complete blood counts. Dim 1 and Dim 2 correspond to the main directions in the multidimensional space that explain the most significant differences in the expression profiles of the markers evaluated in the study. This means that the samples that are more separated along Dim 1 and Dim 2 exhibit more distinct immunophenotypic expression patterns, allowing the identification of subgroups of patients with similar characteristics. These two dimensions represent the primary components capturing the most significant variability in the dataset, enabling the differentiation of patient subgroups. Welch’s *t*-test (unequal variance *t*-test) was applied to evaluate key variables between the defined prognostic and ploidy groups, with statistical significance set at *p* ≤ 0.05 (95% confidence interval).

### 2.6. Correlation Analysis

We conducted a detailed correlation analysis using the corrplot package in R to explore the relationships among various clinical and biological variables across different patient groups. The primary objective was to identify significant correlations specific to particular prognostic categories or ploidy classifications. To achieve this, we analyzed correlations between variables such as immunophenotypic marker expression, cell-cycle phase distribution, and blood cell counts.

Correlation coefficients were calculated using Pearson’s correlation method, with significance levels determined based on a threshold of R ≥ 0.7 for strong positive correlations and R ≤ −0.7 for strong negative correlations. These thresholds allowed us to focus on the most relevant associations. The analysis also included visual representations of the correlation matrices to facilitate the identification of patterns unique to each prognostic group.

## 3. Results

### 3.1. Characteristics of the Study Population

The study included 64 pediatric patients diagnosed with B-cell acute lymphoblastic leukemia (B-ALL) between 2019 and 2020. Of these, 38 were male (59.4%) and 26 were female (40.6%), with ages ranging from 11 months to 14 years. The distribution of patients according to the DNA index (DNAi) was as follows: 31.2% (n = 20) were hyperploid, 46.9% (n = 30) were normoploid, and 21.9% (n = 14) were hypoploid.

### 3.2. Clustering Based on Principal Component Analysis

Principal Component Analysis was used to visualize the variability and clustering of pediatric patients with B-cell acute lymphoblastic leukemia (B-ALL) according to their prognosis, which is closely linked to ploidy status. As shown in Figure 1, the patients clustered into three main categories: those with a good prognosis (hyperploid) are predominantly grouped in the lower-left region (blue), patients with a standard prognosis (normoploid) are concentrated in the central region (red), and patients with a poor prognosis (hypoploid) are spread across the upper region (yellow).

The ellipses surrounding each group reflect the variance within each category, and their clear separation suggests significant differentiation between patient groups. This indicates that PCA is effective in distinguishing different prognosis groups based on ploidy status. The clustering pattern highlights that the underlying characteristics of leukemic cells, such as DNA index and immunophenotypic features, play a key role in patient classification.

### 3.3. Cell-Cycle Alterations

Through Principal Component Analysis (PCA), we identified specific immunophenotypic markers and cell-cycle parameters that significantly differentiated individuals based on ploidy classifications. These parameters were determined by analyzing the variance in the dataset, with the PCA highlighting those that contributed most to the separation of the prognostic groups. A significant finding was the difference in the percentage of cells in the S phase of the cell cycle across the prognostic groups. Specifically, patients with a good prognosis (hyperploid) exhibited a significantly higher percentage of cells in the S phase compared to those with a standard prognosis (normoploid) and those with a poor prognosis (hypoploid) (Figure 2B). The data revealed that good-prognosis patients had an average of 30.06% ± 7.34 of their cells in the S phase, whereas standard-prognosis patients had 6.87% ± 7.25, and poor-prognosis patients showed a markedly lower percentage of only 1% ± 0.17.

### 3.4. Immunophenotypic Variations

Principal Component Analysis (PCA) identified several immunophenotypic markers as relevant for differentiating prognostic groups in B-ALL patients. However, among all the markers identified, HLA-DR was the only one that showed statistically significant differences between the groups. Specifically, patients with a good prognosis (hyperploid) exhibited a lower proportion of HLA-DR-positive cells (30.75% ± 18.23) compared to those with a standard prognosis (normoploid) (52.24% ± 34.26) and those with a poor prognosis (hypoploid) (44.52% ± 53.21) (Figure 2A).

Figure 2A illustrates these differences in HLA-DR expression, clearly showing how the proportion of HLA-DR-positive cells varies across the prognostic categories. This finding highlights the importance of HLA-DR as a crucial component in the immunophenotypic stratification of B-ALL patients.

### 3.5. Global Correlation Analysis

A comprehensive correlation analysis was conducted to explore the relationships among immunophenotypic markers, leukocyte counts, and cell-cycle phases across different patient groups categorized by prognosis: good, standard, and poor. The analysis revealed the diverse correlation patterns that significantly differed between these groups. Specifically, in patients with a good prognosis, stronger positive correlations (R ≥ 0.70) were observed between certain immunophenotypic markers and cell-cycle phases than in those to those with standard or poor prognosis. For example, in the good-prognosis group (Figure 3A), a notable positive correlation was identified between CD19 expression and the S-phase percentage. In contrast, these correlations were less pronounced or absent in the standard- (Figure 3C) and poor-prognosis groups (Figure 3B). 

These distinct correlation patterns highlight the differential behavior of immunophenotypic markers and their potential role in influencing prognosis. The figure panels provide a visual representation of these correlations, with positive correlations indicated by intensifying blue hues and negative correlations by increasing red tones. Parameters without significant correlations are depicted in white.

### 3.6. Detailed Correlation Analysis by Prognostic Group

Following our comprehensive correlation analysis, we conducted a detailed examination of the specific correlations within each prognostic group. In the good-prognosis group, significant correlations were identified, notably between CD34+ and CD22+ cells (R = 0.89) and between lymphocyte and monocyte counts (R = −0.82) (Figure 4A). These findings highlight key relationships that may influence the favorable outcomes observed in this group.

In contrast, the poor-prognosis group exhibited different significant correlations. Specifically, strong correlations were found between CD45low leukocytes and blast counts (R = 0.89), and between CD45low leukocytes and granulocyte counts (R = −0.88) (Figure 4B). These correlations suggest distinct biological interactions that might contribute to the poorer outcomes associated with this group. 

Interestingly, the standard prognosis group did not exhibit any statistically significant correlations (Figure 4C), indicating a more heterogeneous or less defined interaction between the analyzed parameters. These distinct correlation patterns across the different prognostic groups emphasize the unique biological interactions present in each group, potentially contributing to their respective clinical outcomes.

## 4. Discussion

This study focused on the immunophenotypic characterization and cell-cycle analysis of B-cell acute lymphoblastic leukemia (B-ALL) patients, utilizing flow cytometry-based immunophenotyping combined with risk stratification derived from the DNA index (DNAi). Our findings indicate that there are significant differences in immunophenotypic profiles and cell-cycle phases among the prognostic groups defined by the DNAi: good prognosis (hyperploid), standard prognosis (normoploid), and poor prognosis (hypoploid).

We observed a hypodiploidy rate of 21.9%, which is significantly higher than the 5–8% range reported in the literature [11,12]. This discrepancy could be attributed to variations in sample populations or potential regional genetic differences, as well as the specific inclusion criteria applied in our study, such as the exclusion of patients with pre-existing genetic syndromes like Down syndrome or those with particular genetic mutations. Hypodiploidy has consistently been associated with poor prognosis in pediatric ALL, underscoring the importance of accurate detection and the need for further investigation to better understand its impact on risk stratification and disease management.

Our analysis also focused on contrasting specific parameters within immunophenotypes, white blood cell counts, and cell-cycle phases across the prognostic groups determined by the DNAi. To effectively conduct this analysis, we employed Principal Component Analysis, a widely used statistical tool in biomedical research that simplifies complex variables into more manageable categories without a significant loss of information.

The PCA visualizations revealed distinct but partially overlapping ellipses for each prognostic group, reflecting the variance within these pediatric patient populations (Figure 1). This overlap suggests that while there are shared characteristics among the groups, there are also variations in immunophenotypic attributes, blood cell counts, and cell-cycle phases that justify their classification into separate prognostic categories. These findings align with previous studies that have highlighted the complex interplay between immunophenotypic markers and cell-cycle dynamics in B-ALL, underscoring the heterogeneity within prognostic groups [11,12].

Our results reinforce the notion that B-ALL is a highly heterogeneous disease, where even within broadly defined prognostic categories, there can be significant variation in biological markers. This partial overlap observed in the PCA could reflect the underlying genetic and molecular diversity among patients, which may influence their response to treatment and overall prognosis. Further studies are necessary to explore whether these overlapping characteristics could be leveraged to refine prognostic models, potentially leading to more personalized therapeutic approaches.

The PCA revealed significant differences in the percentages of S-phase cells among the prognostic groups. When comparing these groups, we observed that individuals with a good prognosis exhibited higher percentages of S-phase cells. Notably, patients with a poor prognosis displayed a significantly lower percentage of S-phase cells compared to those with a standard or good prognosis (Figure 2B). This marked difference highlights the distinct proliferative activity among the prognostic categories, with poor prognosis patients showing reduced cell proliferation.

This finding is consistent with previous studies that have shown that a higher percentage of S-phase cells is associated with a more favorable prognosis in leukemia and other solid tumors [13,14,15,16]. In leukemia, the correlation between a higher percentage of S-phase cells and better prognosis, as well as enhanced treatment response, has been documented in the literature [17,18]. Conversely, individuals with a poor prognosis display markedly lower percentages of S-phase cells, which might be associated with their adverse outcomes.

Although the mechanism underlying the association between an increased percentage of S-phase cells and a favorable leukemia prognosis remains elusive, research suggests that cells with abundant genetic material generally exhibit extended S-phase transit times [13,14]. This phenomenon may increase their susceptibility to treatments such as vincristine, methotrexate, and 6-mercaptopurine, thereby improving treatment responsiveness and overall prognosis [19,20,21].

The classification of individuals with a poor prognosis, characterized by the smallest percentages of S-phase cells, suggests they might have fewer cells responsive to these treatments. This observation is particularly concerning, given that many recently developed drugs target DNA-synthesizing cells [22,23]. This fact further underscores the importance of considering poor-prognosis individuals as a higher-risk group, as previously suggested [11,24]. Nevertheless, a comprehensive understanding of this subgroup necessitates more in-depth analysis, such as tracking disease progression.

Our findings suggest that cell-cycle regulation, particularly the transition into the S phase, plays a critical role in the prognosis of treatment-naïve patients with acute lymphoblastic leukemia. The literature demonstrates that most ALL cells progressing into the S phase have an undisturbed G1-S transition, with well-regulated cyclin A expression and CDK2 activity [25]. This regulation is essential for effective progression through the S and G2 phases. In our study, patients with a good prognosis exhibited a higher percentage of cells in the S phase, suggesting more efficient cell-cycle regulation and greater susceptibility to chemotherapy targeting DNA synthesis.

In contrast, patients with a poor prognosis displayed a significantly lower percentage of S-phase cells. This could be related to alterations in cyclin A regulation or a reduced activation of cyclin-dependent kinases, which limit their ability to enter and progress through the S phase [26]. Moreover, the lack of activation of the p53 pathway, as highlighted in previous studies, may contribute to decreased proliferation and resistance to apoptosis in these leukemic cells, reducing treatment efficacy [27]. Resistance to apoptosis and the inability to adequately progress through the S phase could be key factors underlying the poorer prognosis observed in these patients.

These observations and their implications need to be confirmed with studies involving larger sample sizes to validate these findings and reinforce the evidence supporting the use of the DNA index and immunophenotypic markers in predicting the prognosis of B-ALL patients. This presents an intriguing area for future research.

The PCA identified several immunophenotypic markers as relevant for differentiating patient groups, with HLA-DR expression emerging as the only marker showing statistically significant differences between prognostic categories. Specifically, patients with a good prognosis exhibited lower HLA-DR expression (30.75% ± 18.23) compared to those with standard (52.24% ± 34.26) and poor prognosis (44.52% ± 53.21).

Previous studies have reported that higher-risk ALL patients often have an increased percentage of HLA-DR-expressing cells, while lower-risk patients show fewer HLA-DR-positive cells, with standard-risk patients falling in between these two extremes [28]. Our findings align with these reports, showing that patients with a good prognosis have the lowest percentage of HLA-DR-positive cells. However, patients with poor prognosis in our study did not show a significant difference in HLA-DR expression compared to the other groups. This discrepancy could be due to sample size limitations or population-specific factors, highlighting the need for further research to validate these findings in a larger cohort.

Additionally, changes in HLA-DR expression have been associated with apoptotic processes in leukemic cells, as shown in other studies [29,30]. The lower HLA-DR expression observed in good-prognosis patients in our study may indicate underlying apoptotic activity or alterations in cell differentiation pathways. These findings underscore the complexity of HLA-DR’s role in leukemogenesis and its potential impact on patient prognosis. Future studies should include larger sample sizes within each prognostic category to enhance the robustness of the PCA results and explore the detailed mechanisms underlying the prognostic significance of HLA-DR expression in B-ALL patients.

To complement our analysis of individual parameters, we conducted a comprehensive correlation analysis between various immunophenotypic markers, blood cell counts, and cell-cycle phases within each prognostic group: good prognosis, standard prognosis, and poor prognosis. This analysis aimed to identify significant correlations that could provide additional insights into the interactions between these parameters within each group, helping to better understand the biological characteristics that differentiate these groups.

In our effort to better understand the interactions among the various parameters evaluated, we identified a significant correlation between monocytes and lymphocytes in patients with a good prognosis, a relationship that was not observed in the other prognostic groups. This correlation, commonly measured through the lymphocyte/monocyte ratio (LMR), is a well-documented biomarker in cardiovascular diseases and solid tumors. In these contexts, lower LMR values have consistently been associated with poorer survival outcomes [31,32].

In studies on cardiovascular diseases and solid cancers, the LMR has proven to be a robust prognostic indicator, reflecting the balance between immune response and systemic inflammation. However, its application in hematologic malignancies such as B-ALL is not as well established. The literature shows that an elevated LMR has been associated with poorer clinical outcomes in other hematologic malignancies, such as diffuse large B-cell lymphoma, Hodgkin’s lymphoma, multiple myeloma, and chronic lymphocytic leukemia (CLL) [33,34]. However, its role in B-ALL has not been well documented, with a clear gap in research exploring its relevance in this context. Our study is among the first to investigate the relationship between LMR and prognosis in B-ALL patients, contributing a novel and potentially significant aspect to the use of LMR as a biomarker in this disease.

Our results suggest that although poor-prognosis patients tended to have lower LMR values, these differences did not reach statistical significance when compared to the good- and standard-prognosis groups. This lack of statistical significance could be due to various factors, including sample size, the intrinsic heterogeneity of B-ALL subgroups, or differences in the underlying biology of the disease compared to other cancer types.

The potential relevance of the LMR in B-ALL should not be dismissed, despite the current findings. It is possible that with more detailed analysis and an approach that considers the interaction of multiple biomarkers, subgroups of patients could be identified where the LMR holds significant prognostic value. Indeed, future research could benefit from integrating the LMR with other immunophenotypic and genetic parameters to develop a more accurate and specific prognostic model for B-ALL.

Following the analysis of relationships between different cell populations, we identified a particularly relevant correlation between the percentages of CD34+ and CD22+ cells in patients with a good prognosis. This correlation suggests the possible co-expression of CD22 on CD34+ blasts, which could have important implications for both prognosis and treatment in these patients. The co-expression of these markers may indicate a subpopulation of leukemic cells that, due to their immunophenotypic profile, could respond better to targeted therapies [35].

CD22 is a marker that is expressed early during B-cell ontogeny and is present in the majority of B-ALL blasts. On the other hand, CD34 is a marker associated with hematopoietic stem cells. The observed correlation between these two markers in the good prognosis subgroup could indicate that these leukemic cells exhibit a more primitive phenotype, making them more susceptible to targeted treatments, such as anti-CD22 monoclonal antibodies and CAR-T cell therapies targeting this marker [36,37].

The significance of this co-expression is notable, as treatments targeting CD22, such as inotuzumab ozogamicin, have been highly effective, achieving complete remissions in a large proportion of patients with relapsed or refractory B-ALL [37]. However, it is crucial to confirm these findings in future studies to establish whether this co-expression is consistently observed in other prognostic groups and to explore the molecular mechanisms underlying this association. Additionally, evaluating the clinical outcomes of good-prognosis patients with CD34+CD22+ co-expression in response to CD22-targeted therapies could provide valuable insights for improving risk stratification and therapeutic strategies [37].

Integrating these immunophenotypic markers into prognostic models could not only enhance the ability to predict outcomes, but also guide the development of more personalized and effective therapies for B-ALL patients.

In individuals with poor prognosis, we identified a significant positive correlation between CD45low leukocyte counts and blast counts (Figure 4B). CD45 is a crucial leukocyte antigen involved in cell differentiation, found across normal and malignant lymphohematopoietic cells, including myeloid, T-cell, and B-cell lineages. During typical B-cell maturation, CD45 expression increases and remains stable in mature cells; however, its reduced presence is notably observed in blasts in B-ALL, which may be associated with an incomplete differentiation state [34].

The correlation observed in our study, specific to poor-prognosis patients, suggests that most of their blasts could be under-differentiated CD45low cells. This under-differentiation aligns with reports in the literature where less differentiated leukemias are often correlated with poorer clinical outcomes [35,36]. This finding highlights the importance of CD45 not only as a marker of differentiation, but also as a prognostic indicator, particularly in the context of B-ALL.

Previous studies have indicated that pediatric patients with less differentiated leukemias tend to have a less favorable treatment response and a higher relapse rate [37]. The low expression of CD45 in blasts might reflect a disruption in normal B-cell maturation, contributing to a more aggressive clinical behavior. This sub-differentiated profile could make these cells less susceptible to conventional therapies, underscoring the need for more targeted therapeutic approaches for this subgroup of patients.

To better understand the implications of this observation, future research should focus on the detailed characterization of these CD45low cells in poor-prognosis patients. Additionally, it would be essential to explore how these characteristics affect the response to current therapies and whether they could serve as a target for new therapeutic strategies. This could include studies on how modulating CD45 expression might influence cell differentiation and potentially improve clinical outcomes [34].

## 5. Conclusions

Our study provides valuable insights into pediatric B-cell acute lymphoblastic leukemia (B-ALL) in pediatric patients by combining DNA index analysis with immunophenotyping to identify patterns that correlate with disease prognosis. Unlike previous studies, we systematically investigated the relationship between the DNA index and the expression of markers such as CD34 and CD22, finding that patients with a good prognosis displayed a notably higher co-expression of these markers. This suggests that the combination of the DNA index and immunophenotyping can play an important role in risk stratification and in identifying subgroups of patients who may benefit from more targeted therapies.

Furthermore, our analysis of the lymphocyte/monocyte ratio (LMR) contributes to the current understanding, as this study is among the first to explore the relevance of the LMR in pediatric B-ALL. Although we did not find statistically significant differences, our results suggest that the LMR could be a valuable prognostic biomarker if studied in larger cohorts or in combination with other immunophenotypic and genetic parameters.

Overall, these findings not only provide new insights into the biology of pediatric B-ALL, but also have the potential to contribute to the development of more accurate prognostic models and to guide more effective treatment strategies. The integration of the DNA index, immunophenotyping, and the LMR represents a comprehensive approach that could improve the risk stratification and clinical management of this disease. However, we acknowledge that including a larger patient population in future studies would be essential to validate and strengthen our conclusions, as well as to further explore the applicability of these findings across different patient subgroups.

## Figures and Tables

**Figure 1 cancers-16-03585-f001:**
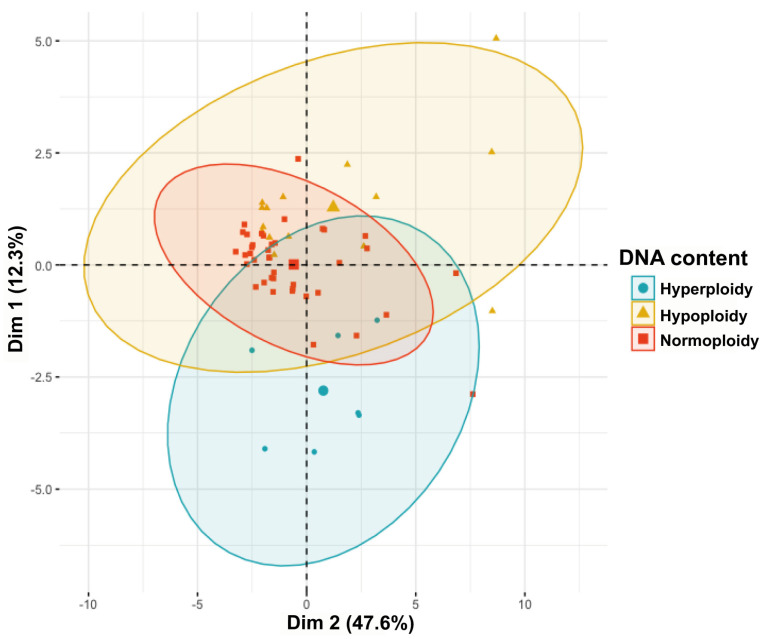
Principal Component Analysis (PCA) highlighting the clustering of B-ALL patients based on prognosis. Patients with a good prognosis (blue, hyperploid), standard prognosis (red, normoploid), and poor prognosis (yellow, hypoploid) form distinct groups with clear separation, indicating significant differences in the underlying characteristics between these groups. The axes Dim 1 and Dim 2 represent the first two principal components obtained from the PCA, capturing the most significant variation in the expression patterns of immunophenotypic markers among the patient samples.

**Figure 2 cancers-16-03585-f002:**
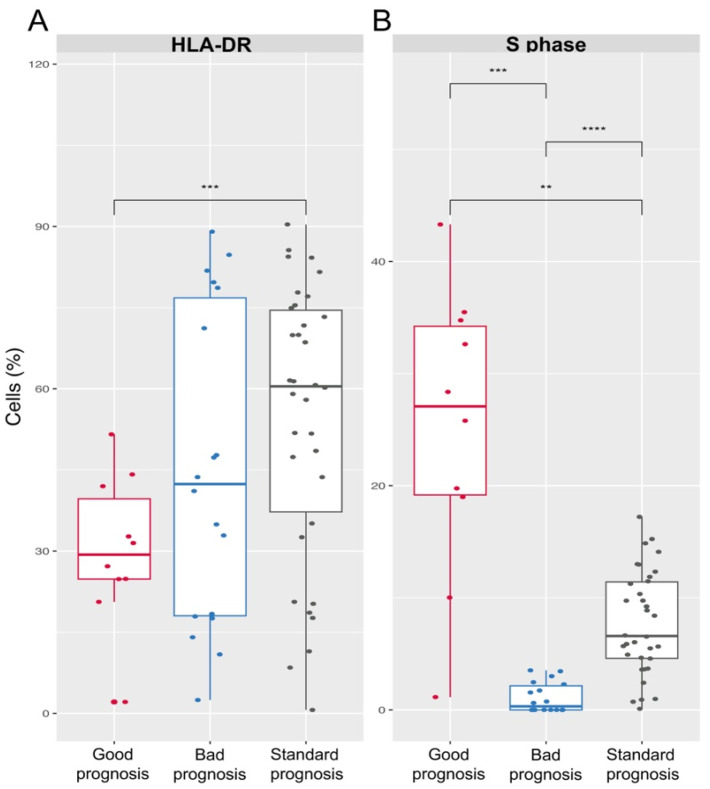
Comparative analysis of cell percentages based on prognostic groups. Differential representation of cell percentages among (**A**) prognostic groups showcasing HLA-DR-expressing cells, and (**B**) prognostic groups highlighting cells in the S phase of the cell cycle. Good prognosis patients (hyperploid) are shown in red, poor prognosis patients (hypoploid) are shown in blue, and standard prognosis patients (normoploid) are shown in gray. Significance levels (** *p* ≤ 0.01, *** *p* ≤ 0.001, **** *p* ≤ 0.0001) were determined via Welch’s *t*-test. Vertical lines depict the standard error. All group comparisons were performed, but only statistically significant differences are shown.

**Figure 3 cancers-16-03585-f003:**
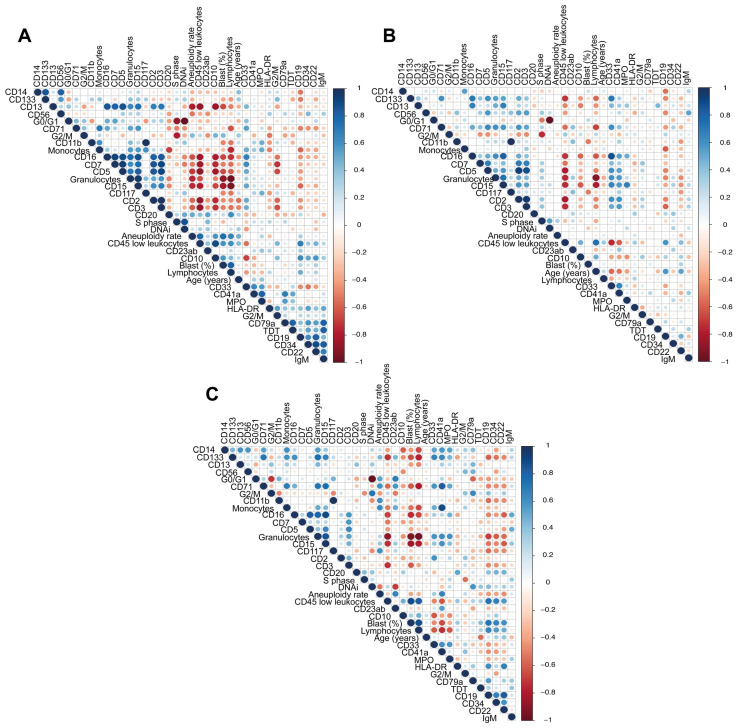
Immunophenotypic, blood cell count, and cell-cycle parameter correlations. A comprehensive correlation analysis displaying the relationships among (**A**) good-prognosis patients, (**B**) poor-prognosis patients, and (**C**) standard-prognosis patients. Direct correlations are depicted by intensifying blue hues, while inverse correlations are indicated by increasing red tones. Parameters without significant correlations are shown in white. The color spectrum representing correlation intensities is provided on the right side of each panel.

**Figure 4 cancers-16-03585-f004:**
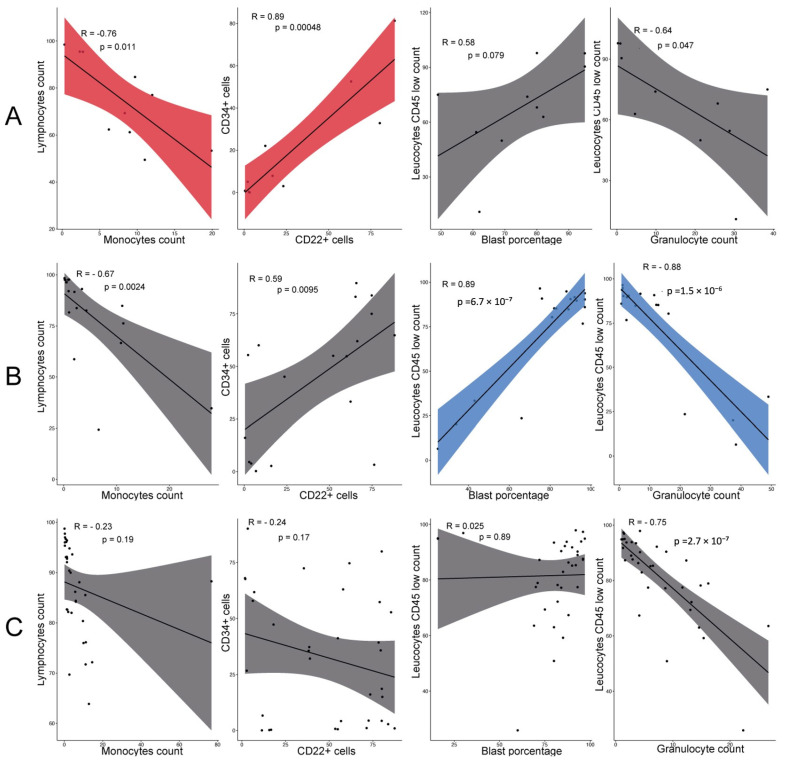
Specific cellular correlations based on prognosis. Analysis of cell population relationships among (**A**) good-prognosis patients, showing significant correlations between lymphocyte and monocyte counts (negative correlation, red) and between CD34+ and CD22+ cells (positive correlation, red); (**B**) poor-prognosis patients, showing significant correlations between CD45low leukocytes and blast counts (positive correlation, blue) and between CD45low leukocytes and granulocyte counts (negative correlation, blue); and (**C**) standard-prognosis patients, where no statistically significant correlations were observed. Statistically significant correlations are defined as R ≥ 0.70 or R ≤ −0.70.

## Data Availability

Dataset available on request from the authors.

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
