# Peer review of "Distinct Immunophenotypes in the DNA Index-Based Stratification of Pediatric B-Cell Acute Lymphoblastic Leukemia"

_cancers, 2024, doi:10.3390/cancers16213585_

Round 1
Reviewer 1 Report
Comments and Suggestions for Authors
The manuscript by Campos-Aguilar et al. entitled: “Distinct Immunophenotypes in the DNA Index-Based Stratification of B-cell Acute Lymphoblastic Leukemia” reports on 64 pediatric patients with B-cell acute lymphoblastic leukemia classified in three groups according to DNA index of the leukemic cells.
The study is aimed to explore:
a) the association of specific immunophenotypic markers with each ploidy group
b) the linked of these associations with prognostic categories
This is a retrospective study conducted on a limited number of cases. The study highlights findings mostly documented in literature, however it also has some point of interest as the correlation between the percentages of CD34+ and CD22+ cells in patients with a good prognosis and the low HLA-DR expression in the same group of patients.
The main weakness of the study concerns the lack of the cytogenetic analysis to validate data resulting from the determination of the DNA index. The percentages of hypodiploid and diploid cases in present study, 21% and 46,9% appear substantially different than those reported in the literature (~ 5% and ~70-80% respectively) for correspondent groups.
Other minor revision concern
Minor points concern:
a) in the title should be specified that the study concerns pediatric patients
b) in the materials CD45 is missing among the markers used for the immunophenotype
c) in the results the percentage of hypodiploid cases is 21.9% while in the discussion the percentage reported is 15%
Author Response
Main Comment 1:
“The main weakness of the study concerns the lack of the cytogenetic analysis to validate data resulting from the determination of the DNA index.”
Response to Main Comment 1:
Scientific literature has consistently demonstrated that the DNA index measured by flow cytometry is a valid and reliable technique that strongly correlates with the karyotype for evaluating ploidy in pediatric acute lymphoblastic leukemia. For example, one study showed a very strong statistical correlation (R = 0.987) between the DNA index measured by flow cytometry and the theoretical index calculated from the karyotype (Rachieru-Sourisseau et al., 2010), while another work found a significant correlation (p = 0.009) between the DNA index and the modal chromosome number determined by karyotype (Forestier et al,.1998). These findings support that the DNA index is a reliable method and, in some cases, more sensitive than the karyotype in detecting aneuploid clones.
Nevertheless, we acknowledge that the inclusion of cytogenetic analysis would have enriched the findings of our study and provided additional valuable data. Unfortunately, in the healthcare institutions in Mexico where this study was conducted, karyotyping is not routinely performed in patients with ALL, which prevented its inclusion in our analysis. As an additional effort to overcome this limitation, the DNA index results generated in this study were donated to the hospital for inclusion in the patients’ clinical records to support clinical decision-making.
References:
- Rachieru‐Sourisseau, P., Baranger, L., Dastugue, N., Robert, A., Geneviève, F., Kuhlein, E., & Chassevent, A. (2010). DNA Index in childhood acute lymphoblastic leukaemia: a karyotypic method to validate the flow cytometric measurement. International Journal of Laboratory Hematology, 32(3), 288-298.
- Forestier, E., Holmgren, G., & Roos, G. (1998). Flow cytometric DNA index and karyotype in childhood lymphoblastic leukemia. Analytical Cellular Pathology, 17(3), 145-156.
Main Comment 2:
“The percentages of hypodiploid and diploid cases in the present study, 21% and 46.9%, appear substantially different than those reported in the literature (~5% and ~70-80%, respectively) for corresponding groups.”
Response:
The variability in ploidy percentages reported in the literature it has been considerable throughout different works. For example, hypodiploid cases have been reported to range from 3% (Yu et al., 2020) to 6.8% (Raimondi et al., 2003), while other studies that focused exclusively on hypodiploid cases reported specific subtype percentages, such as 40.3% for near-haploid and 21% for low-hypodiploid (Holmfeldt et al., 2013). Additionally, Pui et al. 2003 identified 14.7% of hypodiploid cases. In terms of diploid cases, the percentages vary widely, with Raimondi et al., 2003 reporting 93.2% of normoploid cases. This range of variability indicates that substantial differences exist across studies, potentially reflecting variations in populations and classification methods.
Moreover, the difference observed in our study could also be related to the inclusion criteria we applied, such as the exclusion of patients with pre-existing genetic syndromes, like Down syndrome, and those with specific genetic mutations. To clarify this possible explanation, we have added a paragraph in lines 335 to 339 of the manuscript, addressing how these population, genetic variations, and the inclusion criteria used in our study could contribute to the differences observed in ploidy percentages.
References:
- Yu, C. H., Lin, T. K., Jou, S. T., Lin, C. Y., Lin, K. H., Lu, M. Y., ... & Yang, Y. L. (2020). MLPA and DNA index improve the molecular diagnosis of childhood B-cell acute lymphoblastic leukemia. Scientific reports, 10(1), 11501.
- Raimondi, S. C., Zhou, Y., Mathew, S., Shurtleff, S. A., Sandlund, J. T., Rivera, G. K., ... & Pui, C. H. (2003). Reassessment of the prognostic significance of hypodiploidy in pediatric patients with acute lymphoblastic leukemia. Cancer, 98(12), 2715-2722.
- H Holmfeldt, L., Wei, L., Diaz-Flores, E., Walsh, M., Zhang, J., Ding, L., ... & Mullighan, C. G. (2013). The genomic landscape of hypodiploid acute lymphoblastic leukemia. Nature genetics, 45(3), 242-252.
- Pui, C. H., Rebora, P., Schrappe, M., Attarbaschi, A., Baruchel, A., Basso, G., ... & Ponte di Legno Childhood ALL Working Group. (2019). Outcome of children with hypodiploid acute lymphoblastic leukemia: a retrospective multinational study. Journal of Clinical Oncology, 37(10), 770-779.
Minor Comment 1:
“In the title, it should be specified that the study concerns pediatric patients.”
Response:
We have modified the title of the manuscript to specify that the study focuses on pediatric patients. The updated title is “Distinct Immunophenotypes in the DNA Index-Based Stratification of Pediatric B-cell Acute Lymphoblastic Leukemia.”
Minor Comment 2:
“In the materials section, CD45 is missing among the markers used for immunophenotyping.”
Response:
We have revised and modified the “Materials and Methods” section to include CD45 as part of the panel used for immunophenotyping. This correction is now reflected in lines 124 to 127 of the manuscript.
Minor Comment 3:
“In the results, the percentage of hypodiploid cases is 21.9%, while in the discussion, the reported percentage is 15%.”
Response:
We appreciate the observation regarding this inconsistency. We have corrected the hypodiploid percentage, and the correct value of 21.9% has been updated in the discussion, now reflected in lines 334 to 339 of the manuscript.
Reviewer 2 Report
Comments and Suggestions for Authors
1. Materials and methods - stratification of patient samples into prognostic categories – how did you decide on this cut off, please provide the references regarding that matter.
2. line 107 is not clear - “no known genetic or cytogenetic alterations at the time of diagnosis” – you excluded all the patients with genetic or cytogenetic (aneuploidy?) abnormalities? I suppose you meant abnormalities before onset of leukemia?
3. Fig 1 – explain Dim 1 and Dim 2 in figure legends and/or in MM text
4. Fig 2 – what about the MFI of HLA-DR and other markers? This is important since it is known that hyperploid ALL cells show higher (eg. CD20, CD22, CD58…) or lower (CD45) intensity of some markers.
5. Figure 3 – As I can see “CD19 - S-phase” correlation is shown in white (no correlation)?
6. Line 278-280 – how did you calculate the percentage of monocytes and lymphocytes – according to the immunophenotype or only by gating ly+mo on FSC/SSC? Again, what about the MFI of CD22 and CD34. What was the percentage of patients that were CD22+ and CD34+ in your study?
7. Line 285 “strong correlations were found between CD45low leukocytes and blast counts (R = 0.89)” – aren’t CD45 low leukocytes in fact blasts in leukemia? Please be clearer whether you refer to percentage of CD45+ cells or MFI of CD45.
8. Line 344-357 – please elaborate more thoroughly. Do apoptosis, TP53 or other cell cycle regulative pathways play a role in this phenomenon?
9. Line 415 – is this a first report of ly/mo ratio?
10. Highlight the novelty of your research in discussion/conclusion section
Comments on the Quality of English LanguageMinor editing of English language required.
Author Response
Comment 1:
“Materials and methods - stratification of patient samples into prognostic categories – how did you decide on this cut off, please provide the references regarding that matter.”
Response:
The cutoff for stratifying the DNA index (DI) was selected based on previous studies that established its significance in pediatric acute lymphoblastic leukemia. To support this decision, we have added a reference to the work by Yu et al. (2020), which demonstrates that a DI ≥ 1.16 is associated with a favorable prognosis, while a DI < 1.00 correlates with hypodiploidy, providing a clear risk stratification. This justification has been included in the “Materials and Methods” section and can be found in lines 147 to 148 of the manuscript.
Comment 2:
“Line 107 is not clear - ‘no known genetic or cytogenetic alterations at the time of diagnosis’ – you excluded all the patients with genetic or cytogenetic (aneuploidy?) abnormalities? I suppose you meant abnormalities before onset of leukemia?”
Response:
Thank you for pointing out this potential confusion. We have modified the phrase to clarify that patients with pre-existing genetic syndromes, such as Down syndrome, or other conditions involving chromosomal number changes before the diagnosis of leukemia were excluded. This modification can now be found in lines 113 to 117 of the manuscript.
Comment 3:
“Fig 1 – explain Dim 1 and Dim 2 in figure legends and/or in MM text.”
Response:
We have incorporated an explanation of “Dim 1” and “Dim 2” both in the Figure 1 legend and in the “Materials and Methods” section to clarify their meaning. The Figure 1 legend now includes an explanation in lines 225 to 228, indicating that Dim 1 and Dim 2 represent the first two principal components obtained from the Principal Component Analysis (PCA), which capture the most significant variation in the expression patterns of immunophenotypic markers. Additionally, a paragraph has been added to the “Materials and Methods” section, lines 163 to 177, detailing how Dim 1 and Dim 2 correspond to the main directions in the multidimensional space that explain the most significant differences in the expression profiles of the markers evaluated in the study.
Comment 4:
“Fig 2 – what about the MFI of HLA-DR and other markers? This is important since it is known that hyperploid ALL cells show higher (e.g., CD20, CD22, CD58…) or lower (CD45) intensity of some markers.”
Response:
We appreciate the suggestion to consider the Mean Fluorescence Intensity (MFI) for the evaluated markers. However, we decided not to include MFI in our main analysis due to concerns regarding the variability and reproducibility of this measurement. As detailed by Mizrahi et al. (2018), MFI can be highly variable due to factors such as cytometer settings, sample handling, and equipment calibration, with coefficients of variation (CV%) ranging from 7% to 33%. This variability can hinder data comparability and lead to inconsistent interpretations if rigorous standardization procedures are not followed.
Furthermore, using MFI as a quantitative tool requires strict calibration and quality controls, which may not always be feasible in retrospective clinical studies. For these reasons, we chose to evaluate the presence or absence of markers as a more reliable and consistent measure, minimizing the technical variability that could affect the interpretation of our results.
Reference:
Mizrahi, O., Ish Shalom, E., Baniyash, M., & Klieger, Y. (2018). Quantitative flow cytometry: concerns and recommendations in clinic and research. Cytometry Part B: Clinical Cytometry, 94(2), 211-218.
Comment 5:
“Figure 3 – As I can see “CD19 - S-phase” correlation is shown in white (no correlation)?”
Response:
Thank you for your observation. In Figure 3, there are varying degrees of correlation between CD19 and cells in the S-phase across the different panels. Specifically, in Panel A (good prognosis), there is a negative correlation of -0.3, while in Panel B (poor prognosis), the correlation is -0.2. In Panel C (standard prognosis), the correlation is 0.1. Although these correlations are low and not statistically significant”, particularly in the case of patients with a standard prognosis. We appreciate your observation and hope this clarification is helpful.
Comment 6:
“Line 278-280 – how did you calculate the percentage of monocytes and lymphocytes – according to the immunophenotype or only by gating ly+mo on FSC/SSC? Again, what about the MFI of CD22 and CD34? What was the percentage of patients that were CD22+ and CD34+ in your study?”
Response:
Thank you for your observations. The determination of lymphocyte and monocyte percentages was performed using a Cell-Dyn EMERALD Hematology Analyzer (Abbott), as now indicated in lines 123 to 124 of the manuscript. This equipment provided the absolute counts of the different cell populations present in the bone marrow samples.
Regarding the Mean Fluorescence Intensity (MFI) of the CD22 and CD34 markers, as explained in a previous response, the decision not to include MFI in the analysis was based on the technical limitations and inherent variability of this measure, which can be influenced by equipment calibration and other experimental variables. Lastly, in our study, approximately 74% of patients were CD22+ and CD34+.
Comment 7:
“Line 285 ‘strong correlations were found between CD45low leukocytes and blast counts (R = 0.89)’ – aren’t CD45 low leukocytes in fact blasts in leukemia? Please be clearer whether you refer to percentage of CD45+ cells or MFI of CD45.”
Response:
Thank you for your observation. In the context of acute lymphoblastic leukemia (ALL), low expression of the CD45 marker (CD45low) has been identified as a hallmark of leukemic blasts. Previous studies, such as that of Cario et al. (2014), have shown that CD45 expression is related to prognosis and leukemic cell burden in ALL patients. In particular, patients with poor prognosis tend to have a higher proportion of CD45low cells, reflecting a higher blast burden.
Furthermore, the study by Balasubramanian et al. (2021) supports the observation that low CD45 expression is linked to disease aggressiveness and poor clinical outcomes in ALL. In our study, the correlation between CD45low cells and blast count was found to be especially significant in the poor prognosis group of patients, with a correlation coefficient of R = 0.89. This finding is consistent with the idea that low CD45 expression is a reliable marker for the presence and burden of leukemic blasts.
It should be clarified that, in our analysis, this correlation refers to the percentage of CD45low cells and not the Mean Fluorescence Intensity (MFI) of CD45. The relationship observed in our study reinforces the role of the CD45low marker in the assessment of blast burden in ALL patients and its relevance in the prognostic stratification of the disease, in line with previously reported findings (Cario et al., 2014; Balasubramanian et al., 2021).
References:
- Cario, G., Rhein, P., Mitlöhner, R., Zimmermann, M., Bandapalli, O. R., Romey, R., Moericke, A., Ludwig, W. D., Ratei, R., Muckenthaler, M. U., Kulozik, A. E., Schrappe, M., Stanulla, M., & Karawajew, L. (2014). High CD45 surface expression determines relapse risk in children with precursor B-cell and T-cell acute lymphoblastic leukemia treated according to the ALL-BFM 2000 protocol. Haematologica, 99(1), 103-110. https://doi.org/10.3324/haematol.2013.090225 .
- Balasubramanian, P., Singh, J., Verma, D., Kumar, R., Bakhshi, S., Tanwar, P., Singh, A. R., & Chopra, A. (2021). Prognostic significance of CD45 antigen expression in pediatric acute lymphoblastic leukemia. Blood Cells, Molecules and Diseases, 89, 102562. https://doi.org/10.1016/j.bcmd.2021.102562 .
Comment 8:
Line 344-357 – please elaborate more thoroughly. Do apoptosis, TP53 or other cell cycle regulative pathways play a role in this phenomenon?
Response:
Thank you for your observation and suggestion. We have expanded the discussion section to provide a more detailed explanation of the role of apoptosis, TP53, and other cell cycle regulatory pathways in the observed relationship between the S phase and prognosis in patients with acute lymphoblastic leukemia (ALL). Based on previous studies analyzing cell cycle transition and cyclin A expression in ALL, as well as the function of p53 in cell cycle regulation, we have added two paragraphs (lines 398-414) that discuss how the dysregulation of these pathways may contribute to the lower proliferation and poorer prognosis in patients with a low percentage of cells in the S phase.
In these paragraphs, we highlight that alterations in the activity of cyclins and cyclin-dependent kinases (CDKs), as well as the lack of activation of the p53 pathway, could explain the lower proportion of cells in the S phase and the treatment resistance seen in poor prognosis patients. This addition complements our findings and provides a broader context on how cell cycle regulation can influence treatment response and prognosis in ALL patients.
Comment 9:
Line 415 – is this a first report of ly/mo ratio?
Response:
Thank you for your observation. We have included an additional paragraph in the discussion section (lines 465-479) that addresses the relevance of the lymphocyte/monocyte ratio (LMR) in the context of hematologic malignancies. The literature shows that, while the LMR has been used as a prognostic biomarker in other hematologic malignancies such as diffuse large B-cell lymphoma, Hodgkin’s lymphoma, multiple myeloma, and chronic lymphocytic leukemia (CLL), its application in B-cell acute lymphoblastic leukemia (B-ALL) has not been well documented. Our study is one of the first to explore the relationship between LMR and prognosis in B-ALL patients, providing a novel and potentially significant aspect to the use of LMR in this disease.
Comment 10:
Highlight the novelty of your research in discussion/conclusion section
Response:
Thank you for your observation. We have revised and modified the discussion and conclusion sections to more clearly highlight the novelty and contribution of our work. In particular, we have emphasized how our study is among the first to investigate the relationship between the DNA index, immunophenotyping, and the lymphocyte/monocyte ratio (LMR) in pediatric patients with B-cell acute lymphoblastic leukemia (B-ALL). Additionally, we have included perspectives on how our findings may contribute to the development of more accurate prognostic models and the importance of including a larger patient population in future studies to validate and strengthen our conclusions. The revised conclusion is now located in lines 546 to 566 of the manuscript.
Reviewer 3 Report
Comments and Suggestions for Authors
In this manuscript, the authors explore whether specific immunophenotypic markers are associated with each DNAi-based group and their potential connection to prognostic categories. Firstly, the authors utilized flow cytometry to analyze immunophenotypic markers and combined this with DNA index (DNAi) measurements to stratify B-ALL pediatric patients into distinct risk categories. The results showed that hypoploid B-ALL patients displayed a significantly lower percentage of cells in the S-phase of the cell cycle compared to normoploid and hyperploid groups. In addition, distinct immunophenotypic profiles were observed in hypoploid patients, characterized by higher expression levels of HLA-DR and a notable co-expression of CD34 and CD22. Overall, the authors present a interesting result. However, I think some issues should be addressed before publication in Cancers
Some specific comments:
1. In Figure 2A, please add the p-value between good prognosis and bad prognosis.
2. In figure 3, whether the authors can list the specific difference among the different group?
3. In this manuscript, whether the authors can further validate the difference among different groups based on the transcription level or protein expression level?
Author Response
Comment 1:
In Figure 2A, please add the p-value between good prognosis and bad prognosis.
Response:
Thank you for your observation. The p-values for the comparisons between the groups were calculated in the original analysis, but they were not included in the figure as not all differences were statistically significant. In the specific case of the comparison between the good prognosis and poor prognosis groups for HLA-DR expression, the p-value was 0.0913, indicating that it did not reach statistical significance. We have updated the figure 2A legend (lines 265 to 266) to clarify that statistical comparisons were performed between all groups, but only statistically significant differences are shown.
Comment 2:
In figure 3, whether the authors can list the specific difference among the different group?
Response:
We appreciate your observation regarding Figure 3. We carefully reviewed the correlations among the different risk groups and identified the most relevant and unique correlations for each group of patients: good prognosis, poor prognosis, and standard risk.
For the good prognosis group (hyperdiploid), in addition to the correlations presented in Figure 4, which include the negative correlation between lymphocyte and monocyte counts (R = -0.76) and the positive correlation between CD34+ and CD22+ (R = 0.89), we also found additional correlations such as Lymphocytes vs Granulocytes (-0.9589), Granulocytes vs CD15 (0.8039), CD71 vs CD15 (0.8210), CD71 vs CD41a (0.8140), and CD15 vs Lymphocytes (-0.8566). These correlations reinforce the importance of immunophenotypic markers in stratifying the prognosis in this group of patients.
In the poor prognosis group (hypodiploid), besides the correlations shown in Figure 4, such as the positive correlation between CD45low and blast percentage (R = 0.89) and the negative correlation between CD45low and granulocyte count (R = -0.88), additional correlations included: CD15 vs CD16 (0.9385), CD3 vs CD2 (0.9897), Lymphocytes vs Granulocytes (-0.9730), CD33 vs Granulocytes (0.9598), Granulocytes vs CD71 (0.8361), CD15 vs Granulocytes (0.8082), CD45low leukocytes vs Granulocytes (-0.8615), CD45low leukocytes vs CD2 (-0.8549), CD45low leukocytes vs CD3 (-0.8741), Lymphocytes vs CD71 (-0.8341), CD33 vs CD71 (0.8016), CD33 vs CD16 (0.8107), and CD33 vs CD45low leukocytes (-0.8557).
Finally, for the standard risk group (normoploid), the most important and exclusive correlations included: CD5 vs CD7 (0.9435), CD2 vs CD7 (0.9671), CD2 vs CD5 (0.9809), CD45low leukocytes vs CD7 (-0.9248), CD45low leukocytes vs CD5 (-0.9135), CD45low leukocytes vs Granulocytes (-0.9458), CD45low leukocytes vs CD2 (-0.9499), CD10 vs Granulocytes (-0.9134), CD10 vs CD45low leukocytes (0.9173), Lymphocytes vs Granulocytes (-0.9605), and CD41a vs CD33 (0.9099). These correlations reflect the diversity of the immunophenotypic profiles characterizing patients in this group.
Comment 3:
In this manuscript, whether the authors can further validate the difference among different groups based on the transcription level or protein expression level?
Response:
We appreciate your suggestion regarding the validation of differences among groups based on transcription or protein expression levels. Unfortunately, in this study, it was not possible to perform these additional analyses, as the samples were collected and processed exclusively for the analyses presented in our work, and we do not have additional samples stored for further validation. Moreover, since the patients have continued with their respective treatments and their disease status has changed since the time of collection, obtaining new samples is not feasible.
However, we value your suggestion and recognize the importance of conducting transcriptional and protein-level analyses to validate and complement our findings. In future studies, we plan to collect and store additional samples, allowing for complementary analyses that will enrich our understanding of the differences among the various risk groups.
Round 2
Reviewer 1 Report
Comments and Suggestions for Authors
The Authors modified the manuscript as suggested.
Unfortunately, the main weakness of the study, the absence of cytogenetic analisys of the patients useful to support the results of DNA index, can't be overcomed.
Author Response
We appreciate your observations and understand your concern regarding the absence of cytogenetic analysis to support the DNA index results. We would like to highlight that the determination of ploidy through flow cytometry (DNA index) has been widely validated in the scientific literature, demonstrating a high concordance with the results obtained by conventional cytogenetics, such as karyotyping.
Previous studies have shown a strong correlation between the DNA index measured by flow cytometry and the number of chromosomes determined by cytogenetics. For example, Rachieru-Sourisseau et al. (2010) found a very strong correlation (R = 0.987) between the DNA index and the theoretical index calculated from the karyotype, while Forestier et al. (1998) reported a significant correlation (p = 0.009) between the DNA index and the modal chromosome number. These findings support the reliability of the DNA index as an equivalent method to karyotyping for evaluating ploidy in patients with acute lymphoblastic leukemia.
Moreover, flow cytometry is a faster and more cost-effective technique compared to conventional cytogenetic analysis, allowing for more timely and efficient clinical decision-making, especially in resource-limited settings such as the hospitals where our study was conducted. This methodology is widely used in clinical practice for risk stratification and appropriate patient management.
Despite the mentioned limitation, we believe that our results provide valuable information on the immunophenotypic differences among prognostic groups based on the DNA index in pediatric patients with B-cell ALL. We believe that sharing these findings with the scientific community will contribute to the understanding of the disease and can serve as a basis for future research that includes complementary cytogenetic analyses and expands knowledge in this area.
We appreciate your understanding once again and hope that you consider the relevance of our study for the scientific community and clinical practice.
References:
- Rachieru-Sourisseau, P., Baranger, L., Dastugue, N., Robert, A., Geneviève, F., Kuhlein, E., & Chassevent, A. (2010). DNA Index in childhood acute lymphoblastic leukaemia: a karyotypic method to validate the flow cytometric measurement. International Journal of Laboratory Hematology, 32(3), 288–298.
- Forestier, E., Holmgren, G., & Roos, G. (1998). Flow cytometric DNA index and karyotype in childhood lymphoblastic leukemia. Analytical Cellular Pathology, 17(3), 145–156.
Reviewer 2 Report
Comments and Suggestions for Authors
I do not see the added paragraph with TP53, CDK etc. (comment 8) in discussion section.
Please, all the changes made on the previous version of the manuscript should be highlighted (eg. in yellow.)
Comments on the Quality of English LanguageMinor editing of English language required.
Author Response
Comment 1:
I do not see the added paragraph with TP53, CDK etc. (comment 8) in discussion section.
Response:
We apologize for the confusion. The paragraph discussing TP53, cyclin-dependent kinases (CDKs), and cell cycle regulatory pathways was included in the previous version; however, we did not highlight the changes made, which may have led to it being overlooked. We have corrected this oversight and have ensured that all changes, including this paragraph in the Discussion section (lines 398 to 414), are properly highlighted to facilitate your review. Thank you for bringing this to our attention, and we apologize for any inconvenience caused.
Comment 2:
Please, all the changes made on the previous version of the manuscript should be highlighted (eg. in yellow).
Response:
We understand the importance of clearly indicating the revisions made to the manuscript. In the revised version, we have highlighted all changes made since the previous submission in yellow, as per your request. We hope this will facilitate your review of the revised manuscript.
Comments on the Quality of English Language:
Minor editing of English language required.
Response:
We have conducted a thorough review of the manuscript to address the language issues noted. The modifications made to improve the English have been highlighted in green in the revised manuscript. We believe these adjustments enhance the clarity and readability of the text, effectively communicating our findings to the readers.